# A qualitative examination of the experiences and perspectives of interprofessional primary health care teams in the distribution of the COVID-19 vaccination in Ontario, Canada

**Rachelle Ashcroft**[1]*, **Catherine Donnelly**[2], **Peter Sheffield**[1], **Simon Lam**[1], **Connor Kemp**[3], **Keith Adamson**[1], **Judith Belle Brown**[4]

1 Factor-Inwentash Faculty of Social Work, University of Toronto, Toronto, Ontario, Canada, 2 Faculty of Health Sciences, Queen's University, Kingston, Ontario, Canada, 3 Frontenac, Lennox, and Addington Ontario Health Team, Kingston, Canada, 4 Department of Family Medicine, Schulich School of Medicine and Dentistry, Western University, London, Ontario, Canada

* rachelle.ashcroft@utoronto.ca

**Data Availability Statement:** Relevant data may not be publicly shared as it contains information that could potentially be used to identify

## Abstract

### Background

Primary health care (PHC) teams contributed to all phases of the COVID-19 vaccination distribution. However, there has been criticism for not fully utilizing the expertise and infrastructure of PHC teams for vaccination distribution. Our study sought to understand the role PHC teams had in the distribution of the COVID-19 vaccine in Ontario, Canada. The key objective informing this study was to explore the experiences and perspectives of interprofessional PHC teams in the distribution of COVID-19 vaccination across Ontario.

### Methods

A qualitative approach was used for this study, which involved 39 participants from the six health regions of the province. Eight focus groups were conducted with a range of interprofessional healthcare providers, administrators, and staff working in PHC teams across Ontario. The sample reflected a diverse range of clinical, administrative, and leadership roles in PHC. Focus groups were audio-recorded and transcribed, while transcriptions were then analyzed using thematic analysis.

### Results

We identified the following four themes in the data: i) PHC teams know their patients; ii) mobilizing team capacity for vaccination, iii) intersectoral collaborations, and iv) operational challenges.

### Conclusions

PHC teams were an instrumental component in supporting COVID-19 vaccinations in Ontario. The involvement of PHC in future vaccination efforts is key but requires additional

participants. Responses to questions contain information describing Family Health Team communities, practice settings, and providers' personal roles and responsibilities, some of whom live in small rural communities where such information could be used to identify individuals. Data requests can be sent to Dr. David Brennan, Associate Dean of Research, at david. brennan@utoronto.ca.

**Funding:** This study was funded by the Pre-Tenure Award from the Factor-Inwentash Faculty of Social Work at the University of Toronto. The funders played no role in study design, data collection and analysis, decision to publish, or preparation of the manuscript.

**Competing interests:** The authors have declared that no competing interests exist.

resourcing and inclusion of PHC in decision-making. This will ensure provider well-being and maintain collaborations established during COVID-19 vaccination.

# Background

The World Health Organization (WHO) declared a Public Health Emergency of International Concern in January 2020 due to the spread of the novel coronavirus, COVID-19 [1]. By the end of 2020, Pfizer/BioNTech, Oxford and AstraZeneca reported the development of safe and effective vaccinations for COVID-19 [2–4]. Effective and efficient distribution of a vaccine is essential for the international recovery from the COVID-19 pandemic [5–8]. Leadership in delivering vaccinations can be found within primary health care (PHC) teams due to their expertise in administering regular vaccines such as influenza, and the ability to bring together the skills of family physicians and/or nurse practitioners working in partnership with a range of interprofessional healthcare providers [8, 9]. However, over the course of the COVID-19 vaccine distribution, there has been criticism for not fully engaging PHC and maximizing the expertise of the teams during both the vaccination rollout planning and distribution processes of the COVID-19 vaccine [9, 10].

## Role of PHC in the distribution of the COVID-19 vaccination

Although variations exist across countries, jurisdictions, and organizations, PHC has contributed to all phases of the COVID-19 vaccination rollout [2, 11]. As the first point of access in the healthcare system, PHC has a long history of successfully delivering immunization programmes [12–18]. This success has been built on a system that centres on counselling and a vaccine infrastructure, as exemplified by the delivery of millions of influenza immunizations every year and the longstanding operationalizing of childhood vaccination programmes [8, 9].

Given such historical expertise and experience, there are several benefits to engaging PHC in the COVID-19 vaccination distributions [8]. PHC has established deep connections and long-standing relationships formed over time with patients and communities, which can be an asset in ensuring vaccination rollouts minimize inequalities and reduce access barriers for patients [8, 9]. PHC's expertise in health promotion activities, patient education, and comprehensive knowledge of patients and communities may also be an asset in overcoming vaccine hesitancy related to vaccinations [8].

Despite the history and success of PHC in delivering vaccinations, the first phase of vaccine distribution in Canada revolved around mass vaccination programs operated by public health, hospitals, and community pharmacies [9]. In subsequent phases, PHC's involvement in COVID-19 vaccination distribution planning was limited in some jurisdictions, resulting in a lack of knowledge that could have assisted in more effective delivery of the vaccine [9, 10].

## Study rationale

While there is some emerging data on the experiences of family physicians with COVID-19 vaccinations [8], there are few studies that have explored the role of interprofessional PHC teams in planning and delivering COVID-19 vaccines [11]. Early reports demonstrate that PHC teams have contributed to all phases of the COVID-19 vaccination rollouts [2, 11], yet little is known about the unique experiences of interprofessional PHC teams in the distribution of the COVID-19 vaccine in Ontario, Canada. The key objective informing this study was to explore the experiences and perspectives of interprofessional PHC teams in the distribution of

COVID-19 vaccination across Ontario, Canada. Knowledge gleaned from this study can provide critical insights to inform policy and practice decisions about how PHC teams can support future COVID-19 boosters and other vaccinations.

## Methods

### Design

Our study used a qualitative description approach, using focus groups to explore the experiences and perspectives of 39 participants. Qualitative description is an approach ideally suited to describing and exploring health care and gaining insights into a phenomenon that has not been well described [18]. Given the novelty of the COVID-19 vaccination rollout, this approach fits the context of the research and its aims. The research team included a range of research, clinical, and leadership expertise with disciplinary training and backgrounds spanning social work, rehabilitation sciences, family medicine, and PHC health services research. The research was carried out in partnership with the Association of Family Health Teams of Ontario (AFHTO), a provincial organization in Ontario that provides leadership and advocates on behalf of PHC teams. Ethics approval was obtained from the University of Toronto Research Ethics Board (REB Protocol #41679). This research involving human participants was conducted in accordance with the Declaration of Helsinki.

### Setting

As Canada's most populous province with 14.7 million residents [19], Ontario began its COVID-19 vaccination program on December 14, 2020 [20–22]. Located in Ontario, Family Health Teams (FHTs) are Canada's largest team-based model of PHC [22]. FHTs are a type of patient medical home [23] which integrates physical, mental, and other health services by bringing together a cadre of interprofessional healthcare providers working in partnerships with family physicians and/or nurse practitioners. At the time of our study, there were 184 FHTs [24] operating in Ontario with each team differing in size, services, and providers, although most teams were comprised of family physicians, nurse practitioners, nurses, social workers, pharmacists, dietitians, and other providers including occupational therapists and physical therapists [22, 25].

### Sample and recruitment

A purposive sampling technique was used to recruit eligible participants, which included healthcare providers of any disciplinary background, administrators, and other staff working in PHC teams who contributed to the COVID-19 vaccination distribution. We also sought to recruit participants from the six health regions in the province including West, Central, Toronto, East, North East, and North West [26], as well as seeking participants from both rural and urban locations. Our community partner at AFHTO assisted with recruitment by sending emails about the study to executive directors in FHTs who then shared the study information within their teams. AFHTO also sent emails to AFHTO's community of practice groups comprised of a range of interprofessional providers and promoted the study on their website and in their newsletter. We also promoted the study on social media (i.e., Twitter). The recruitment email included an information sheet that was approved by the REB and it explained the study goal, methods, and the name and contact information of the principal investigator (RA) and project coordinator (SL). Potential participants then contacted the project coordinator by email to express interest in participating in the study.

## Data collection

Data was collected using virtual focus groups that were conducted with an online video platform. A semi-structured interview guide, informed by the literature and developed by our team, guided the focus group interviews (see S1 File). Focus groups were selected as the method because they foster interaction amongst participants, which can then elicit a deeper understanding of participant's experiences [27, 28]. When possible, focus groups were comprised of participants with similar disciplinary backgrounds. Participants were sent a copy of the consent form by email and were asked to return a signed copy of the consent form to the researchers by email before commencement of the focus group. Each focus group was co-facilitated by two research team members (RA/CD/SL), audio-recorded, and then transcribed verbatim to generate a transcript for analysis. Focus groups spanned between 60–90 minutes in length. Participants were randomly assigned a code to promote confidentiality. Field notes were made immediately following each focus group.

## Data analysis

Data collection and data analysis took place simultaneously, with data analysis following the thematic analysis process as outlined by Braun and Clark [29]. The six steps as outlined by Braun and Clarke [29] included i) familiarization with the data, ii) generating initial codes, iii) searching for themes, iv) reviewing the themes, v) defining and naming the themes, and vi) producing a report. Codes and themes were iteratively derived from the data. One researcher (PS) was assigned as primary data analyst with two other researchers (RA/SL) serving as secondary data analysts. The primary and secondary data analysts familiarized themselves by reading transcripts prior to coding. The primary and secondary data analysts met weekly to discuss data and emerging codes, leading to the creation of initial codes. Initial codes were then grouped into themes. A data analysis sub-committee comprised of the data analysts and two additional research team members with expertise in qualitative research (CD/JBB) reviewed the initial codes and assisted in naming and defining themes during two virtual meetings. Analysis was completed once data saturation was reached. All members of the research team then met to discuss the themes and offered insights on the interpretation of the data. NVivo12 was used for organizing data analysis.

## Results

A total of 39 participants engaged in eight focus groups. Five of the focus groups were provider-specific, and three were comprised of different interprofessional healthcare providers (IHPs) when we did not have enough participants for a provider-specific focus group. The average size of the focus groups was five participants. Participants reflected geographic diversity across each of the five health regions: West (n = 10), East (n = 9), North East (n = 6), Central (n = 5), Toronto (n = 5), and North West (n = 4). Participants were a mix of urban (n = 21), rural (n = 16), and mixed (n = 1) settings, with n = 1 unknown. Table 1 provides an overview of each of the 8 focus groups, while Table 2 provides details of the professional roles represented by study participants.

The following four themes were identified in the data: i) PHC teams know their patients; ii) mobilizing team capacity for vaccination, iii) intersectoral collaborations, and iv) operational challenges. Table 3 provides an overview of the four themes, as well as the accompanying sub-themes.

**Table 1. Overview of the composition of eight focus groups.**

| Focus Group # | Types of Providers | Number of Participants |
|---|---|---|
| 1 | Executive Directors | 6 |
| 2 | Quality Improvement Decision Support Specialists | 5 |
| 3 | Nurses | 5 |
| 4 | Nurses | 4 |
| 5 | IHPs* | 5 |
| 6 | IHPs** | 4 |
| 7 | IHPs*** | 5 |
| 8 | Family Physicians | 5 |

*included Clerical Staff, Health Promoter, Pharmacist, and Respiratory Therapist

**included Clerical Staff and Nurses

***included Dietitian, Nurse Practitioner, and Pharmacist

## 1. PHC teams know their patients

**Trusting relationships.** Across all focus groups, providers stated that PHC understands the unique needs of patients because of the strong trusting relationships. The nurse focus group explained, "*Given that patients trust their primary care provider. . .most of them express they'd rather get their vaccines with their primary care people rather than. . .having to go to mass clinics that where they don't know anybody*" (FG3, P1, Nurse Practitioner). The importance of the relationship was highlighted: "*The patients that did come, they were thrilled that they came to their clinic where they felt comfortable, where the children, the parents knew us and felt comfortable*" (FG4, P4, Nurse Practitioner). Similarly, one of the IHP focus groups elaborated on how the relationship facilitated patient engagement for vaccination: "*There's a relationship that our patients have with their family doctors. They come to us for information, they called us for months and months prior to us actually getting our hands on the vaccine. . .just leveraging that relationship and being one of many sites where they can get their vaccine was great*" (FG5, P4, Health Promoter). Having a trusting relationship gave patients a sense of assurance, as described by a participant from the QIDSS focus group: "*Patients were more receptive to receiving the vaccine from someone that they knew, even if. . .they weren't at the office but knowing, 'Oh hey, I recognize this RN giving me this vaccine, I feel a lot more comfortable*" (FG2, P2, QIDSS). Across all focus groups, PHC providers explained that relationships with patients

**Table 2. Professional roles represented by study participants (N = 39).**

| Professional Roles | Number |
|---|---|
| Nurses | 8 |
| Executive Directors | 6 |
| Family Physicians | 5 |
| Quality Improvement Decision Support Specialists | 5 |
| Nurse Practitioners | 4 |
| Pharmacists | 4 |
| Clerical | 3 |
| Health Promoter | 2 |
| Dietitians | 1 |
| Respiratory Therapist | 1 |
| **TOTAL** | **N = 39** |

**Table 3. List of four themes and the related sub-themes.**

| Theme | Sub-Themes |
|---|---|
| PHC teams know their patients | Trusting relationships<br>Patient-centred education<br>Easily reach priority patient populations |
| Mobilizing team capacity for vaccination | Vaccination experts<br>Shifting roles<br>Minimizing vaccine wastage |
| Intersectoral collaborations | Primary care leadership<br>Public health: training and information<br>Hospitals: sharing costs and navigating the red tape<br>Municipalities: space and supplies<br>Inadequate inclusion in system-level planning |
| Operational challenges | Vaccine procurement<br>Financial costs<br>Sustainable human resources<br>Inadequate booking system<br>Inefficient electronic tracking of vaccinations |

facilitated vaccine-related education, even during non-vaccine-related appointments. As explained by a nurse: "*In terms of patients really trusting their primary care providers. . .when patients came into the clinic, when they haven't—they're not even getting their vaccines, they always had questions about COVID-19 vaccinations*" (FG3, P3, Nurse).

**Patient-centred education.** Participants explained that PHC is a central source of information and education on all health matters, and as a result, PHC had the infrastructure for patient education already in place:

> *People in our community are used to coming to the Family Health Team, reaching out to their family doctors, their other health care providers when they're looking for health information. . .They were coming to us and waiting for us to share that information. . .We have a Facebook page, a website, we publish weekly articles in our local newspaper. . .we already had all those things in place, so we just built on those and were able to have a really a successful communication strategy and really keep our population informed when it came to the delivery of vaccines.* (FG5, P5, Health Promoter)

Patient education about the COVID-19 vaccination was uniquely tailored to a patient's health situation, because of the continuous engagement patients have with PHC:

> *We have the patient's chart in front of us. A lot of patients that had those unique issues, like an organ transplant patient, or someone with some blood cancers. . .they were able to come in and have a real discussion about getting their vaccine. . .So we were able to really see their health issues and relate that to the COVID vaccine.* (FG4, P4, Nurse Practitioner)

Furthermore, several PHC teams created educational materials tailored to the specific needs of their patients: "*We actually created our own patient information sheets because we looked at all sorts of ones from the Public Health units. . .and. . .a lot of them are quite confusing, and we know the questions our clients ask, and what sort of information they want, and what sort of details they're not really interested in*" (FG4, P2, Nurse). Patient education was particularly useful in situations when patients were initially vaccine hesitant: "*We could provide some education there and sometimes family members would end up getting the vaccine when they were hesitant to coming in the beginning*" (FG5, P3, Pharmacist). Some participants explained

that their team had even reached out to some patients with a history of mental health concerns to ensure that patients had adequate information and support: "*We identified people who from a mental health perspective wanted additional information or resourcing or support. . .it was related to both COVID and vaccinations. . .we would do sort of these 30 minute calls with people just to help support them on that*" (FG1, P3, Executive Director).

**Can easily reach priority patient populations.** Across all focus groups, participants described proactively reaching out to patients who were higher-risk and/or eligible for the vaccine:

> *I was using the list of the various conditions that were high risk. . .I was. . .was pulling lists of names. . .for physicians to review. . .and phone calls were made out from those lists. . .that's what I was doing, was getting the target group, so that we could approach patients. . .because then we knew that they were in the target group.* (FG2, P5, QIDSS)

The nurse focus group noted, "*When vaccines were first available, we made cold calls to our highest risk patients first and supported appointment booking*" (FG7, P1, Nurse Practitioner). Elderly and seniors were prioritized: "*We have quite a few elderly patients, so we started giving immunizations right in when it was available in April or so*" (FG4, P4, Nurse Practitioner). PHC teams prioritized other higher-risk populations, such as patients living with HIV: "*We tried to do some. . .targeted, HIV-specific kind of groups because there's a high population in one of the clinics*" (FG3, P5, Clinical Nurse Specialist). Several focus groups explained that PHC teams vaccinated children and youth when it became appropriate to do so: "*We ended up giving a number of pediatric vaccines, because our clinic is small and child friendly. We made it really fun, we had some treasure boxes and a less intimidating setting for the small littles*" (FG7, P1, Nurse Practitioner). PHC provided an environment that was physically conducive to the needs of children and youth. A family physician explained:

> *. . .Especially for the children, because we have a clinic that has lots of separate little rooms, and so they all had their own little private space and they didn't get influenced by potentially other negative experiences. . .they really liked that small, more community feel, so it was just a nice way of making sure everybody felt comfortable.* (FG8, P2, Physician)

Tools in Electronic Medical Records (EMRs) provided some PHC teams with the capacity to easily identify and reach priority patients: "*In our clinic. . .you can do a search and figure out how many people are 75 and up kind of a thing, so our patients never book- phone to book, really. What happened is, a front desk phoned and called them*" (FG4, P4, Nurse Practitioner). Participants across all focus groups described the benefits of leveraging the EMR to identify specific patient populations eligible for the COVID-19 vaccination. For instance, the use of the EMR allowed a health promoter in a small community in northern Ontario to identify and contact patients who lived in rural areas and were eligible for vaccination:

> *Being such a small community. . .all of the residents. . .are patients of our Family Health Team. . ..At the beginning when only very specific people were eligible for the vaccine, it was easy enough for us to be able to go through our EMR and be able to see, 'Okay, so these are our patients. . .this is our population that's over 80,' and then kind of start to tackle immunizations that way.* (FG5, P5, Health Promoter)

Additionally, participants explained that PHC teams located in rural communities have a comprehensive understanding of the community's population and as a result, were able to easily contact patients eligible for vaccination.

## 2. Mobilizing team capacity for vaccinations

**Vaccination experts.** Participants overwhelmingly emphasized the vast array of relevant expertise within their PHC teams. All focus groups noted the extensive expertise in planning and operationalizing vaccinations. According to the nurse focus group: "*In primary care we do vaccination every day, it's part of our livelihood*" (FG4, P3, Nurse). This was reflected in one of the focus groups with IHPs which underscored the expertise that PHC has with vaccinations:

> *We've had that previous experience before when it comes to flu immunizations...so we just felt that we were in the best position to be able to take on this project locally...We had previous experience with hosting immunization clinics, like our flu clinics, for example, where we were used to immunizing you know 100, 150 people in an evening clinic...so we just built on that and were able to develop a really efficient vaccination plan for our community* (FG5, P5, Health Promoter)

Various team members assisted with vaccinations. According to a pharmacist, "*anybody who could vaccinate, were vaccinating*" (FG7, P4, Pharmacist). Across focus groups, participants explained how their teams quickly mobilized their capacity and contributed as needed. The physician focus group explained:

> *Our health promoter ended up being the clinical observer...watching to see if anybody was feeling pre-syncopal, he'd run over with juice...this week we're a little light on scribes because we're getting back up and running, and, and I just was talking to our ED of the Family Health Team, and she's going to go over and scribe...it's sort of all hands on deck* (FG8, P3, Physician).

As part of the interprofessional team, pharmacists were another provider type actively involved in the vaccination rollouts: "*Myself and other pharmacists were involved, not only in preparing the vaccine, but also administering vaccines...we got to do a lot of the different roles*" (FG5, P3, Pharmacist). The focus group with executive directors elaborated: "*We're lucky enough to have a full-time pharmacist...she was the clinic lead for our FHT-lead vaccination clinics and also became kind of our knowledge expert.*" (FG1, P4, Executive Director). Nurses working in PHC were instrumental in administering the COVID-19 vaccinations. For example, "*Most of the IHPs [interdisciplinary healthcare providers] were the ones doing the vaccines. I know the physicians were definitely supportive, and they were definitely reviewing those lists to identify patients, but it was really the RNs and NPs and RPNs...giving the vaccines.*" (FG2, P2, QIDSS).

**Shifting roles.** Across focus groups, some participants explained that they shifted roles to accommodate the new demands presented by the COVID-19 vaccination distribution. One of the IHP focus groups indicated, "*My role did change, I've been in the health promoter role for seven years, but for the last two years, my role has really been focusing on vaccine planning, vaccine rollout so I was the lead for that project for the last couple of years*" (FG5, P5, Health Promoter). Participants from the QIDSS focus group similarly explained that their role changed to support the vaccination distribution: "*The COVax training is kind of how I started getting involved...I got trained by Public Health and then we were deployed out...to the different clinics that were...going to be using vaccinations, and then, training them on everything which was a little rough because we had very minimal training. So we were kind of learning as we went too*" (FG2, P4, QIDSS). PHC teams adapted to the new demands: "*Once we got the hang of it, everyone just kind of did everything—from admin, to all those consent forms, we kind of got like a*

*system down so that eventually when we got like Moderna and Pfizer later on, we knew what to do*" (FG3, P3, Nurse).

**Minimizing vaccine wastage.** Across focus groups, participants spoke about PHC teams' meticulous attention and measures to delivering the COVID-19 vaccination with minimal wastage of the vaccine. Teams developed processes for vaccination and conducted team training:

*I also had to also develop. . .videos to help train the nurses. . .because, at the very beginning. . .vaccine availability was not so great, we were trying to squeeze every single drop out of every single vial. . .so sort of looked at how we would train for some of those techniques. And, and the storage, and making sure that all of that was safe: the labelling, making sure that everything was labelled properly* (FG7, P2, Pharmacist)

By drawing on the capacity of their EMR, reaching eligible patients, and booking the appropriate number of patients, PHC teams minimized vaccine wastage which was a priority in the early phases of the vaccination roll-out: "*The front desk were very good at avoiding any waste. I don't think we really had any waste of vaccine at all because we. . .would book the number of people for those number of vials. . .there was a lot less waste [in PHC]*" (FG4, P4, Nurse Practitioner). Similarly, one of the focus groups with IHPs emphasized PHC's ability to minimize wastage:

*We would. . .pull data from our EMR to know who was eligible, and we had staff calling every single one of our patients. . .It was very time-consuming but it made sure that our clinics were full, that we didn't waste doses, and we always had a backup list of patients to call if we had no-shows or cancellations, so it just gave us a lot more control over our clinics, and allowed them to run pretty efficiently.* (FG5, P5, Health Promoter)

### 3. Intersectoral collaboration

**PHC leadership.** Across all focus groups, intersectoral collaboration was identified as a key factor that helped facilitate the vaccination rollouts: "*What worked really well was the collaboration. . ..Because of those connections that we had with the community centers and even the CHCs. . .and being able to coordinate everything through a central area*" (FG7, P2, Pharmacist). Collaborations spanned across PHC, the Ontario Health Teams (OHTs), and other key community organizations, as described by the physician focus group: "*I'm the medical director of the Family Health Team. . .we were sort of liaison with primary care. . .and the OHT, and we just struck a working group*" (FG8, P2, Physician). Intersectoral collaboration was essential for planning and sharing of resources:

*[The] county is so big, no one organization had enough staff to do everything. So we formed COVID strategy table—included Public Health, both of our local hospitals, EMS, the county Family Health Teams and our family physicians. And through that group is how we managed everything COVID. . .vaccines, staffing, clinics. . .Staffing in the retirement homes, outbreak management. We did it as a collective, because none of us had enough staff or capacity to do it on our own, so we came together as a county, and did it as a whole.* (FG6, P4, Nurse)

For some teams, drawing on existing collaborative relationships enabled quick planning for COVID-19 response:

*We have a [City Name] collaborative group. We've been meeting for six years, we're not big enough to be an OHT, but that's every partner—public health, paramedics. . .there's total of*

*three Family Health Teams. . .and there was an Aboriginal Health Center. . .and the hospital. So we in February already had a collaborative meeting and we switched the agenda to be totally emergency response going forward.* (FG1, P5, Executive Director)

Participants emphasized the leadership role that many PHC teams played in the roll-out of the COVID-19 vaccinations, particularly in rural communities: "*I also think geography played a huge part at least where I'm located, where you know the FHT is the only game in town, so there was no other option but for them to be involved, and, and reaching these patients*" (FG2, P2, QIDSS).

**Public health: Training and information.** All focus groups emphasized that collaboration with public health was essential to successful vaccination distribution. Public health provided information that guided planning and decisions on geographical areas to target vaccinations: "*Public Health involvement, for instance. . .postal codes. . .So they made a list of the postal codes with the least vaccination to kind of incentivize us to kind of take a look, and so we kind of worked on that.*" (FG2, P1, QIDSS). Several focus groups explained that collaborations with public health were most helpful when there was direct and immediate contact: "*We were getting scads and scads of data constantly about the timing of the vaccines, and who meets the guidelines for the vaccines. . .If it didn't make sense to me, then I would then stop everything and we'd make a phone call to the Health Unit and talk to the nurse*" (FG4, P4, Nurse Practitioner). The sharing of information with public health was helpful to provide support for staff at any given time:

*The collaboration was super helpful. We didn't have to, as a clinic, determine what information was necessary, there was always new stuff coming out about the vaccines. . .all these things were changing all the time. And so Public Health was able to distill that, and provide to us what we needed to know, when we needed to know it.* (FG6, P2, Clerical Staff)

Focus groups emphasized the importance of the relationships that facilitated the collaborative experiences with public health:

*What worked really well here. . .it's that relationship with Public Health. So maybe it's an advantage of being in a smaller rural community, but there was no bureaucracy. There was. . .definitely a bit of a storming period in the beginning, where we were just trying to figure out everyone's roles. . .But in the end. . .it was extremely efficient and coordinated and Public Health did take the lead, and they were very organized. . . that partnership, I couldn't' say enough good things about.* (FG7, P4, Pharmacist)

For many focus groups, the collaborative relationship between PHC and public health strengthened during the pandemic: "*Prior to COVID. . .I always found Public Health worked in a silo. It was totally separate to anything anyone else did, and the only time you heard from them was when someone had an STD. Now. . .we have more cooperation. . .we're collaborating. . .I'm hoping that will continue and it won't go back to being siloed*" (FG6, P4, Nurse). There was a desire for the collaborative relationship with public health to continue going forward: "*The relationships that we did build, we should continue to foster those. . .we shouldn't go back to the way we were doing it for the last 20, that was just probably inefficient and never got the attention that needed*" (FG8, P5, Physician). However, focus groups noted that not all PHC teams had seamless experiences with public health, as noted in the executive director focus group: "*Honestly, we found that our FHT MDs had greater urgency and ability to move faster, but had to expend lots of energy and hours negotiating with Public Health to move forward*" (FG1, P4, Executive Director).

**Hospitals: Sharing costs and navigating the red tape.** Hospital collaborations facilitated PHC's involvement in vaccination distribution by sharing costs and supplies. The executive director focus group spoke at length about this: "*The beauty of what happened. . .the hospital was able to bill for services, for providing vaccinations and we were able to invoice them for out-of-pocket costs, because we wouldn't have been able to do it without that*" (FG1, P3, Executive Director). The role of the local hospital can also include providing supplies to the vaccination distributions: "*[The] hospital covered supplies that were not supplied by public health for our mass clinics*" (FG1, P6, Executive Director). Hospital collaborations helped navigate the red tape related to financial costs, which enabled PHC's participation and leadership in the vaccination distributions as explained in the physician focus group:

> *We had a horrible moment where funding really didn't look like it was going to be possible at all. It was [Hospital Name]. . .CEO. . .who just was super supportive and just helped us basically push through any red tape. . .so that was what actually allowed us to go ahead, was just having a hospital that was just willing to bust all the red tape for us and just as long as we were able to organize ourselves and do the work and get it up and running, they were going to figure out a way to help with the funding.* (FG8, P3, Physician)

The creative use of budgeting and sharing of resources enabled primary care to provide sufficient staffing for vaccination efforts: "*I sort of posed the question. . .Can we not share some of these resources? And then it was even more magical because what we did was, we had some of the. . .support staff that were hired through the. . .OHT line of funding, and then we would borrow them at the FHT*" (FG8, P2, Physician).

**Municipalities: Space and supplies.** Collaborations with local government, politicians, and community leaders emerged to facilitate access to space and supplies to run vaccination clinics. "*Our municipality, they were excellent, they gave us a lot of resources, like cones that we could use and a tent outside. . . and the arena. . .*" (FG1, P1, Executive Director). Municipal collaborations were integral for finding space: "*[The] community center gave us the space to run our clinic. . .and given the center was closed due to restrictions, it worked out very well. . .The mayor. . .was a huge supporter of our work*" (FG8, P5, Physician). Municipal partnerships were identified as critical in being able to run large vaccination clinics: "*We will continue to vaccinate in clinic but large clinics will need to be partnered affairs and in a larger venue. . .arenas, local high schools*" (FG1, P3, Executive Director).

**Inadequate inclusion of PHC in system-level planning.** While many experienced positive intersectoral collaborations, each focus group expressed concerns about the inadequate inclusion of PHC in system-level planning related to the COVID-19 vaccination distribution.

> *There was a major, major hiccup when the vaccine Task Force for the province did not involve anyone from primary care. Not just doctor–any nurse, pharmacist, they just didn't involve primary care at all, nor did they involve Public Health until several months in. . .if you try to plan something that you then want to roll out to primary care but primary care is not at the table at the planning stage, they always get it wrong. So, so you have to involve us from the beginning.* (FG8, P5, Physician)

PHC was not adequately included in regional planning, as expressed in the focus group with executive directors: "*I was surprised at how disassociated some primary care groups were from the planning with health units—particularly in the larger urban sites where hospitals took the lead and I think that this is actually not a good model. . .the partnered approach with primary care, health units, hospitals actually generates better results*" (FG1, P3, Executive

Director). Similarly, the nurse focus group shared concerns about the oversight of excluding PHC's involvement in regional planning:

> *Working with the Health Unit has been a bit of a challenge. There was no communication back and forth, it was like pulling teeth. . .When it comes to flu vaccination clinics. . ..we've always taken the lead on those things, and then we were completely left out for COVID. We weren't included in any of the discussions. It was a bit frustrating, and it was very difficult to get information.* (FG4, P3, Nurse)

The focus group continued to discuss challenges related to limited communication with public health: "*We're a smaller area and. . .we couldn't get a hold of a nurse at the Health Unit. We didn't even know what was going on at the Health Unit, we didn't know when the vaccines were*" (FG4, P4, Nurse Practitioner). Barriers were discussed in the family physician focus group in addressing how PHC could participate in regional planning:

> *It was just being really bull-headed from our Family Health Team and just insisting that we had a role, and that that we should be a part of it. At first we actually came out in January of 2021 and said, 'We really want to be a part of this. . .this is going to be important,' and our Public Health, unfortunately, at that time, told us that there wasn't a role for us. . .we pushed through that, and sort of insisted that there was. . .an enormous role for us.* (FG8, P3, Physician)

FHTs that wanted to participate in vaccinations had to convince the public health authority in order to do so: "*We kept telling Public Health, like, 'You are not using us to the fullest, please put us to work.' And so there was a lot of just pushing from our perspective, to make it happen*" (FG8, P4, Physician). There was advocacy required for PHC to engage in regional planning:

> *[M]y experience was interesting in the sense that it started off with. . .Public Health not really realizing that primary care wanted to assist them at all in the vaccine rollout. . .They called a meeting. There were at least 30 of us that attended. They were very surprised that we were interested. . .It just showed that there really needs better communication between primary care and Public Health—it's long overdue.* (FG8, P5, Physician)

This led some teams to have frustrations with the exclusion of PHC from the outset, and indicated that several challenges with the roll-out would have been mitigated if PHC's input was engaged from the outset of planning:

> *[I]t frustrates me. . .had we been at the outset, COVax would have been integrated with our EMRs. . .Like we would have been thinking about that stuff ahead of time. And I think the integration of COVax into our EMRs, knowing who was vaccinated, who wasn't, who had had what doses, trying to like recruit people and advertise accordingly, and all of that stuff, was probably one of the biggest headaches, and I feel like—like many things, a lot of that stuff could have been avoided had primary care been at the table sooner. . . And that definitely jaded me. . .like, primary care is the backbone of immunizations across this country, for every-thing else, and why were we sort of left in the dark.* (FG8, P4, Physician)

Along with limited participation in system-level planning, many participants raised chal-lenges that they faced because of the limited communication and information that PHC received about decisions that were made related to the vaccination roll-outs: "*Sometimes it was*

*like the patients were telling us about an update that we didn't even know about, They're saying 'we're eligible,' and we're like, Wow! and then, and then you hear about it after the fact, you kind of felt silly* (FG1, P1, Executive Director). Across several focus groups, vaccination rollouts were seen as uninformed and inadequately prepared to respond to patient questions:

*I remember specifically earlier on. . .the province. . .would announce like certain things, it's the first time I would hear it. . .I would have to watch the news and learn everything at the same time as everyone else is. So all of these patients are expecting us to already have this knowledge and we didn't. So it was kind of hard to just prepare and navigate all of these incoming calls when we didn't even have answers of our own.* (FG3, P3, Nurse)

The difficulties from lack of communication was highlighted in one of IHP focus groups:

*When it came to releasing the information, especially about booster shots. . .they were changing the requirements for them so quickly and we would hear it in the news. We would know we'd be preparing for one set of clinics for a certain demographic, and then within a week it had changed again, so I think having it released on the news is tough on the people that put clinics together.* (FG5, P2, Medical Receptionist)

When PHC teams received different messages from different sectors, communication challenges heightened:

*[W]hen there are too many hands in the pot, it would sometimes get confusing. At the beginning, there was different information coming from public health compared to the Ministry for example, so a lot of discrepancies and messaging that kind of left a lot of work for us in terms of communication, like, 'Okay, so are we following what public health is saying, or the Ministry of Health?', and so just trying to help ourselves, and also our community, with some of that messaging.* (FG5, P5, Health Promoter)

Furthermore, participants explained that without PHC participating in the planning, decisions were not aligned with the capabilities of PHC: "*Primary care needs to be at the table, and there needs to be an understanding of what we are able to do, and what we're not able to do*" (FG8, P3, Physician). Indeed, focus groups described the variations across PHC and need to be considered by system-level planners:

*When these decisions are made. . .they have to keep in mind that not every team is able to. . .provide what's expected. . .depending on what resources they have, what they can tap into. . .So to make these blanket, 'this is what's going to be done,' without knowing actually how the teams are working and what to help, and, and resources they have, I think that makes it hard for a lot of teams that are smaller, and the impact that that has on primary care in general is huge.* (FG3, P2, Nurse)

## 4. Operational challenges

**Vaccine procurement.**   Across all focus groups, participants shared difficulties related to actually obtaining the vaccine itself. Participants described onerous logistical steps that needed to be satisfied before obtaining the vaccine. As noted by in the focus group with executive directors, "*We begged them to be able to have Pfizer on hand. Went through many hoops to be allowed to store vaccine here*" (FG1, P1, Executive Director). The physician focus group described an onerous legal document that accompanied the vaccination:

*If you wanted to do vaccines in your office as a family doctor or nurse practitioner, you had to sign a memorandum of understanding, and you had to have this whole legal documents signed. . .You need legal assistance if you don't have that, you have to pay for it. It was quite a lengthy document. . .they didn't really budge. . .[and] takes up a lot of time we didn't have. So that made things more difficult, unfortunately.* (FG8, P5, Physician)

Several focus groups described difficulties related to the delivery and pick-up of the actual vaccine, as noted by a pharmacist: "*I didn't know whether or not I should go. . .it felt like the logistics were way too complicated, because when we reached out to Public Health, they would redirect us and say. . .you can pick it up. . .Instead of like delivering it to us. . .logistics was really difficult*" (FG7, P2, Pharmacist). For some, vaccine availability inhibited PHC teams' capacity to do vaccinations: "*[W]e ran clinics in the building itself, which were kind of limited because of. . .the availability of vaccine. There were times when we could have given so many more, but. . .the vaccines were going out to the pharmacies before they were coming out to the Family Health Teams*" (FG3, P1, Nurse Practitioner). One of the IHP focus group shared the challenges of obtaining vaccines which inhibited planning:

*Getting vaccine was another challenge. . .which caused a time crunch to schedule clinics. We would tentatively plan a weekend clinic and staff it, but ultimately couldn't book patients until we had vaccine in hand which sometimes wouldn't come until a couple days before the clinic. We had to cancel clinics once or twice when vaccine didn't arrive.* (FG5, P4, Health Promoter)

Procurement of the COVID-19 vaccination varied across PHC teams:

*I'm not sure about everyone else's clinics, but our clinic gets them from a third-party source. . .vaccination supplies is a problem. So sometimes we're scheduled a certain delivery date, but it doesn't come, so that of course impacts our patients. I'm not sure why it's not like the same for my clinic versus other clinics*" (FG3, P3, Nurse).

Some focus groups explained that they would reach out to community pharmacies when vaccination supplies were depleted: "*Sometimes I would even run to some of these community pharmacies and grab a vaccine from them*" (FG7, P2, Pharmacist). The pharmacist went on to explain that vaccination shortages have been a historical challenge even before the COVID-19 pandemic:

*Historically, we were always challenged getting our vaccines from Public Health. Every year, when the flu shot comes out—this year was no different, it was probably worse. . .delays, delays, delays. The RPNs order the vaccine in the clinic, and then they go pick it up. And there's always a list of what's not available in the fridges, all the time. We've probably been without shingle shots now for 90 days. I don't know what the issue is. . .historically, we always seem to have these issues getting vaccines.* (FG7, P5, Pharmacist)

Participants representing northern and rural communities emphasized the unique challenges they encountered obtaining the vaccine. The executive director focus group noted, "*Access to vaccines is also difficult for us just because they come from Thunder Bay. . .they have to get here*" (FG1, P6, Executive Director). Other teams needed to assign someone from their team to drive and pick-up vaccines: "*Transportation of vaccines. . .was also a challenge, especially in winter months. Thunder Bay is 3.5 hours way one way on a good day. . .so we had a volunteer drive. . .every 2 weeks to pick up vaccines*" (FG5, P5, Health Promoter).

**Financial costs.** PHC teams faced challenges with the financial costs associated with the COVID-19 vaccination distribution. Some focus groups noted that the vaccination distribution was costly, as described by the QIDSS focus group: "*Some physicians...only did it for one or two days just to use up the vaccines they had. They found it...cost too much money to do it through [the PHC clinic]*" (FG2, P4, QIDSS). Several focus groups described new costs associated with the vaccine distribution, such as costs related to public awareness campaigns: "*Things like putting it on the radio, putting it on signs, all that kind of stuff again that costs money-...there wasn't money coming specifically to the different teams to communicate that you were willing to give the vaccination to people outside of your own patient rosters*" (FG3, P1, Nurse Practitioner). Similarly, the physician focus group expressed concerns about costs incurred during the COVID-19 vaccination rollouts related to supplies: "*The Family Health Team was saying, like my project manager and nurse manager, 'Are we going to get any support for this? We're ordering all these things, I don't know if we have any more budget'*" (FG8, P2, Physician). The QIDSS focus group stated, "*They're struggling to provide just primary care...let alone all of these duties...passed along to the [Family] Health Teams without additional resources being provided*" (FG2, P3, QIDSS).

Several focus groups emphasized that their PHC team received little additional funds for the human resourcing of vaccination clinics. The physician focus group explained:

*It's interesting to hear the funding side of things...honestly I don't know that there was really any extra funding directly to us, other than what physicians would bill OHIP. Our Family Health Teams' staff would work in some of the mass clinics, but just get paid their salary that they would get paid for from the Family Health Team...there basically was no money—no extra money from anywhere that I'm aware of. So it's fascinating to know how some people got extra money to make a lot of this work and not others.* (FG8, P4, Physician)

The focus group with nurses also expressed similar sentiments about the lack of new funding: "*There was no money. We weren't getting paid. You're on salary, no money was coming into the Family Health Teams*" (FG3, P1, Nurse Practitioner). Similar sentiments were shared in the executive director focus group: "*We didn't bill...my time and that sort of administrative coordination time*" (FG1, P3, Executive Director).

Costs accrued created challenges for some PHC teams: "*We don't have the money to run these clinics*" (FG3, P1, Nurse Practitioner). It was the costs that deterred some PHC teams from participating in the distribution of the COVID-19 vaccination: "*Some primary care teams in our area were hesitant to provide COVID vaccines in primary care as they felt like they were offloading work to them when public health had the resources*" (FG1, P1, Executive Director).

**Sustainable human resources.** All participants spoke about the challenges associated with sustaining adequate human resources for the various vaccination efforts. Where possible, PHC teams brought in additional staff to assist with administrative activities: "*Organizing our own COVID vaccine clinics was just very time-consuming: contacting clients to get them in, and then the whole process of having the check-in, filling out all the stuff, we had to get like separate admin to do that because our admin are already swamped [with] their day to day stuff*" (FG4, P2, Registered Nurse). All focus groups discussed provider exhaustion and burnout. According to one physician, "*Our nurses and admin were sort of getting a bit burnt out because a lot of them were doing this after hours, and on the weekend*" (FG8, P2, Physician). Focus groups noted that the vaccination efforts relied on PHC providers working on top of their regular activities:

*You need to do this vaccination [clinic] at certain times, but at the same time, you still need to do your regular job...you're giving more than 12 hours of your time just to help with the*

*vaccination, but you're expected to because, again as a nurse, you're one of those that can give the vaccine. . .. there are times that I'm working seven days a week.* (FG6, P1, Registered Nurse)

Choices about the types and amounts of providers to engage in the vaccination distribution efforts had to be made in relation to each team's patient care demands and available human resources. For example, one PHC team had nurses involved in the vaccinations with the exception of the nurse practitioners who mainly continued to do routine PHC activities: "*On our teams, [it] was mainly the RNs and RPNs that administered the vaccines, occasionally the NPs, but the NPs are primary care providers in our teams, so they have higher value work to do, to see the chronic patients with chronic conditions*" (FG2, P3, QIDSS). PHC leaders made decisions about how to engage some providers so that they contributed to the vaccination rollouts, but also continued responding to the daily demands of PHC. The executive director focus group noted, "*I didn't want my physicians involved, the involvement that they did was following up with somebody who was actually concerned or had questions*" (FG1, P5, Executive Director). Similar sentiments were noted in the physician focus group, "*At the beginning, some family docs were helping out in the mass clinics. But it's not a very productive use of our time, we found*" (FG8, P1, Physician). Given that the demands for mental health services were high during the pandemic, mental health providers on PHC teams—such as social workers and mental health therapists–minimally participated in direct vaccination distribution efforts. According to the focus group with executive directors, "*The reality was that. . .[there was] such a high demand for social workers in COVID that realistically we had to refocus those people back on their role because they needed to be seeing patients. The demand was so high*" (FG1, P1, Executive Director). Similarly, one of the IHP focus groups noted: "*Our mental health therapists, we left [them] where they were. They were definitely needed in their offices seeing patients*" (FG7, P4, Pharmacist).

All focus groups expressed concerns that the strain on human resources created challenges for providing regular patient care during the vaccination rollouts. "*We're trying to continue running a clinic throughout all of this, right? We're going to see patients. . .we're all checking labs, this is all extra work*" (FG8, P2, Physician). Although providers expressed the importance of vaccination distribution efforts, many focus groups discussed that the vaccination efforts shifted some of the attention away from regular PHC activities, "*It was a strain on resources all around. . .it did take away from some of the regular clinic things, but obviously was absolutely needed*" (FG3, P2, Nurse).

Some focus groups discussed concerns related to the delay of certain types of preventative measures and routine PHC: "*Nursing time was taken away from things like INRs and infant immunizations and pap tests. . .everything that nurses would normally do, there was no additional FTEs provided*" (FG2, P3, QIDSS). The nurse focus group expressed similar concerns, "*That just kind of delayed a little bit in terms of other preventative care that. . .I would focus on. . .a lot of things were delayed*" (FG3, P3, Nurse). The delay in routine care led to a backlog for patients: "*We have so much work to do to catch up on cancer screening, and just getting patients back in the office. Spirometry, reviewing education plans, all of that*" (FG7, P5, Pharmacist). "*While we were focusing a lot on the pandemic, there were other things that got left behind*" (FG8, P5, Physician).

**Inadequate booking systems.** Across all focus groups, participants indicated that many PHC teams struggled with establishing booking processes to accommodate the new demands associated with the COVID-19 vaccinations, and to minimize confusion amongst patients about the various vaccine booking sites that were available: "*People in our community. . .didn't use the provincial booking system, we did our own booking for probably the first year and a half*

of clinics. . .people saw how difficult it was sometimes to actually book an appointment, whether it was online or calling the system" (FG5, P5, Health Promoter). Establishing efficient mechanisms and processes for booking patients for vaccinations was a significant undertaking, as described by the executive director focus group: "*We were creating kind of work-around processes for booking, we set up a telephone line for people that were having a hard time with the provincial booking system, and we were also booking for our internal clinics before we were able to participate in the provincial booking system*" (FG1, P4, Executive Director). Team success in vaccination, according to a QIDSS, was dependent on having adequate IT infrastructure: "*The teams that did the most vaccine is the team that has the best IT infrastructure and it's no coincidence, because. . .when you have patients that are sent that email, 'book online,' just, you print the list in one click, and, and you enter the information in one click, it's just, makes a big difference*" (FG2, P1, QIDSS).

**Inefficient electronic tracking of vaccinations.** All participants spoke about the need for a comprehensive process for tracking COVID-19 vaccinations. As stated in the QIDSS focus group, "*I think we all saw at the beginning of this how important it was to know who got vaccinated and when they got vaccinated, especially when it came to figuring out who should we be calling next, and I think there's a lot more work to be done on that front*" (FG2, P2, QIDSS). A significant challenge that all focus groups noted related to difficulties with vaccination tracking with the EMR and the provincial electronic database for COVID-19 vaccination tracking referred to as COVaxON. At the outset of vaccinations, it was challenging to find a mechanism to consistently track vaccinations:

> *A lot of our job involves making sure the data in the EMR is correct and that was somewhat of a challenge with respect to vaccine data. . .when the mass vaccination clinics and the long-term care clinics first started. . .COVaxON didn't actually even exist initially. So we were having all these legacy reports come in and vaccinations that had been administered once in the past, right, and then we had to ensure the EMR was updated so that when we produced lists of patients who had one, two or three doses, they were actually accurate, which was challenging.* (FG2, P3, QIDSS)

Many focus groups described numerous challenges with the COVaxON system. "*COVax is not user friendly*" (FG4, P1, Nurse) and "*We did have some struggles with COVax*" (FG1, P6, Executive Director) were frequent sentiments raised in focus groups. Similarly, "*COVax was always the hardest piece to all of this. It wasn't giving the vaccines. . .primary care is all used to vaccinating patients, it's running multiple different electronic medical records at the same time, and it was really tricky*" (FG2, P2, QIDSS).

Many participants described separately documenting patient vaccinations in the provincial COVax system as well as the practice-site EMR: "*It was given as COVax in our clinic, [and] it was also documented in your EMR but the two systems don't talk to one another, and that is a drawback*" (FG6, P4, Nurse). Focus groups described the process of inputting vaccinations in COVax as an administrative burden that was onerous and time-consuming: "*The COVax piece was, definitely created more work. We probably could have got through more patients if it was a little more streamlined. . .it definitely took, took time away*" (FG3, P2, Nurse). Because the COVax system was not integrated into the practice-level EMR, many participants explained that it required duplication of tracking: "*We don't have access to like COVax booking, so we still have to contact people, book them in our EMR and double enter everything in COVax is—it's just a huge time burden*" (FG8, P1, Physician). According to the executive director focus group: "*Giving the shot was the easy piece of the process. All the other planning, admin, data entry was the more challenging pieces*" (FG1, P1, Executive Director).

# Discussion

This study provides insights into the experiences and perspectives of interprofessional PHC teams who participated in the distribution of COVID-19 vaccinations and builds on the literature that has examined the experiences of primary care physicians in the COVID-19 vaccination distribution [8, 30]. The strong message that we heard across various providers, leaders, and staff members was that PHC teams were optimally positioned to play a central role in the COVID-19 vaccination distribution. Participants highlighted the relational nature of PHC and the longstanding experience and expertise in delivering vaccinations. As teams, they were able to quickly mobilize and draw on the collective strengths of a breadth of professions in a way that solo primary care physicians may not be able to. One of the values of interprofessional team-based environments is that the range of different types of providers have diverse expertise needed to effectively implement vaccination rollouts. What was also evident was the extent of the variations that existed across PHC teams in terms of their abilities to navigate some of the challenges and obstacles they faced. Some of these variations may be because of differences inherent in the community context, patient population, and/or organizational structures such as team composition, resourcing, and leadership [31, 32]. At a time of international health crisis, the agility by which some PHC teams have been able to identify and contact eligible patients is impressive [33]. Vaccination guidance provided by the National Advisory Committee on Immunization (NACI) in November 2020 did not prioritize key populations for early COVID-19 immunization due to several factors [33]. Despite this lack of guidance, and significant shortcomings in the dissemination of practical vaccination operations protocols, PHC teams were able to determine their own priority populations and facilitate practical items like appointment bookings. PHC teams were also able to quickly shift focus between priority populations when needed. The quick response may have been because of the existing vaccination expertise, and established team processes from past initiatives, such as flu shot clinics.

Findings from this study suggest that PHC teams were well positioned to contribute to the COVID-19 vaccinations due to their comprehensive understanding of patients developed through trusting relationships that have been built over time [34]. This longitudinal relationship is a major advantage of PHC and may mean that patient education initiatives might be more effective in PHC settings because they can be tailored to unique patients' needs and made accessible to equity-deserving patient populations [2]. Pre-existing knowledge of patients' personal health and social history is an advantage that enables PHC to provide patient-centred vaccination education that is most relevant to patients and the surrounding community [2]. Previous reviews of COVID-19 vaccine hesitancy have noted that community-centred and targeted approaches to vaccination are required to reduce hesitancy in minority and migrant populations [35]. Overcoming vaccine hesitancy was not explicitly considered in the design of the COVID-19 vaccination rollouts in most jurisdictions [2] yet may be an asset of administering vaccinations in the context of PHC and should be taken into consideration in future vaccination strategies [2]. Well-established connections between PHC, public health, local hospitals, community health clinics, community agencies, and other community partners, drove the level of engagement and leadership that PHC teams had in vaccination efforts. Such collaborations were essential for accessing space, resources, funding, and knowledge, and overcoming "red tape" that may have limited some teams' capacity for vaccinations. Undoubtedly, PHC teams in rural and Northern regions appear to have provided significant leadership in their communities. Based on our study findings, PHC teams were very motivated to participate yet teams in some jurisdictions seemed to be left out of system-level discussions of vaccination efforts, especially early on. Having to push policy and decision-makers to utilize the resources of PHC represents a system inefficiency. Establishing policies for

emergency response in similar situations could foster early communication between health agencies and PHC. An important finding in our study relates to the importance of establishing and maintaining relationships between PHC and public health to prevent returning to silos once the pandemic subsides, as is the goal of integrated care.

Operational challenges were one of the key hurdles that PHC teams faced in order to participate in the vaccination rollouts. The uncertainty in vaccine supply hampered the operational rollout of vaccination clinics, which suggests that the provincial government may need to examine processes for cold storage and distribution in PHC going forward [1]. Surprisingly, findings in our study support the notion that supply chains across the province were quite heterogeneous, which makes it difficult to maintain consistent and accurate guidance on vaccine procurement. In Northern communities, volunteers driving over 3.5 hours to the nearest city centre to pick-up vaccines is not a sustainable vaccination delivery model. Canadian physicians have expressed frustrations and concerns about the process of Canada's vaccine rollout, including frustrations with poor communication regarding the planning and rollout strategy, and limited consideration for the unique needs of rural and Northern communities [3]. The challenges faced by more rural and Northern teams may have contributed to further inequity in vaccine coverage for marginalized groups in those regions [36, 37]. From this vantage point, our study provides insights needed to strengthen PHC's involvement in the next phase of the COVID-19 pandemic and beyond [38]. To rollout longer-term, more permanent mass vaccination strategies, PHC requires emergency funding, supplies, and resources [36–39]. Demands during the COVID-19 pandemic only heightened the widespread health human resources crisis in PHC [40]. The lack of human resources and staffing issues may also have fueled the notion that some PHC teams were not as actively involved in vaccination efforts. Many PHC teams faced several challenges that restricted their involvement in the rollouts and depleted their financial and human resourcing capacity [41]. Participants emphasized the concerns and confusions that arose related to the online booking of appointments and the lack of clarity on which system to use. Based on our findings, it appears that some PHC teams developed their own solutions in the early stages of vaccine rollout. Our findings suggest that the quality of existing IT infrastructure available to teams appears to have influenced the rate and relative ease with which teams could perform vaccinations. The heterogeneity of IT capacity across PHC teams adds more barriers and may perpetuate further structural inequities across PHC teams [31]. It is also unclear if team IT infrastructure alone drives improvement, as many rural regions may have access to good systems, yet patients also need to have access to appropriate technology to properly interface with those systems. Findings from our study highlight the need for guidance and user-friendly electronic vaccination tracking tools. In our study, electronic documentation was a major barrier to vaccination rollouts in PHC.

All the vaccination efforts resulted in challenges in maintaining regular patient care and detracted from the time required for PHC to complete recommended screening duties and prevention activities [42]. PHC providers and teams were already feeling overwhelmed with regular operations prior to the pandemic, and the massive backlogs in regular screening and care that have resulted from the necessary repurposing of health human resources will be felt long into the future. Greater consideration is needed on how to bolster the support and strength of the team to mitigate the growing rates of provider burnout in PHC.

A limitation is that our study was focused on one type of team-based PHC model in Ontario, Canada, so findings may not be applicable to all PHC settings. Although understanding the experiences of key players in the healthcare system is important, it is important to recognize that the lack of patients' perspectives is a limitation in our study. Healthcare providers in our study overwhelmingly emphasized the importance and role of trusting relationships in the distribution of the COVID-19 vaccination yet without the perspective of patients, we

cannot confirm the accuracy of healthcare providers' perspectives given the dynamic nature of therapeutic relationships. Despite this limitation, we believe our study provides unique insights about the experiences and perceptions of interprofessional providers and PHC teams.

## Conclusion

PHC teams were an instrumental component in supporting COVID-19 vaccinations in Ontario. Because of the longstanding experience in delivering vaccinations, and diverse expertise, PHC teams were able to quickly mobilize and draw on the collective strengths of a breadth of professions. PHC teams need to be included in the planning and strategizing of future vaccination endeavors to optimize their capacity.

## Supporting information

**S1 File. Semi-structured interview guide.**
(DOCX)

## Acknowledgments

We would like to acknowledge the support and involvement of the primary care providers who participated in a focus group. Thank you to Kavita Mehta–past CEO of AFHTO—who made substantial contributions to the conception and design of this work.

## Author Contributions

**Conceptualization:** Rachelle Ashcroft, Catherine Donnelly, Simon Lam, Keith Adamson, Judith Belle Brown.

**Data curation:** Rachelle Ashcroft, Catherine Donnelly.

**Formal analysis:** Rachelle Ashcroft, Catherine Donnelly, Peter Sheffield, Simon Lam, Connor Kemp, Keith Adamson, Judith Belle Brown.

**Funding acquisition:** Rachelle Ashcroft, Keith Adamson, Judith Belle Brown.

**Investigation:** Rachelle Ashcroft, Catherine Donnelly, Peter Sheffield, Simon Lam.

**Methodology:** Rachelle Ashcroft, Catherine Donnelly, Simon Lam, Keith Adamson, Judith Belle Brown.

**Project administration:** Rachelle Ashcroft, Simon Lam.

**Resources:** Rachelle Ashcroft.

**Software:** Rachelle Ashcroft.

**Supervision:** Rachelle Ashcroft, Catherine Donnelly, Simon Lam, Judith Belle Brown.

**Writing – original draft:** Rachelle Ashcroft, Catherine Donnelly, Peter Sheffield, Simon Lam, Connor Kemp, Keith Adamson, Judith Belle Brown.

**Writing – review & editing:** Rachelle Ashcroft, Catherine Donnelly, Peter Sheffield, Simon Lam, Connor Kemp, Keith Adamson, Judith Belle Brown.

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
