## [Decision Letter · Decision Letter 0]

20 Nov 2023

PONE-D-23-23366A Qualitative Examination of the Facilitators and Barriers Shaping Primary Health Care Teams’ Experiences in the Distribution of the COVID-19 Vaccination in Ontario, CanadaPLOS ONE

Dear Dr. Ashcroft,

Thank you for submitting your manuscript to PLOS ONE. After careful consideration, we feel that it has merit but does not fully meet PLOS ONE’s publication criteria as it currently stands. Therefore, we invite you to submit a revised version of the manuscript that addresses the points raised during the review process.

Academic Editor:

The reviewers have provided their comments and recommendations. Based on their feedback, you are encouraged to revise the manuscript, addressing and responding to each comment specifically.

We look forward to receiving your revised manuscript.

Kind regards,

Elif Ulutaş Deniz

Academic Editor

PLOS ONE

Additional Editor Comments:

Dear Rachelle Ashcroft,

Manuscript PONE-D-23-23366 titled "A Qualitative Examination of the Facilitators and Barriers Shaping Primary Health Care Teams’ Experiences in the Distribution of the COVID-19 Vaccination in Ontario, Canada" which you submitted to PLOS ONE, has been reviewed.

The reviewers suggest some major revisions to your manuscript. Therefore, I invite you to respond to the reviewer(s)' comments and revise your manuscript.

Sincerely,

Elif Ulutaş Deniz

Reviewers' comments:

Reviewer's Responses to Questions

**Comments to the Author**

1. Is the manuscript technically sound, and do the data support the conclusions?

Reviewer #1: Yes

Reviewer #2: Yes

2. Has the statistical analysis been performed appropriately and rigorously? 

Reviewer #1: Yes

Reviewer #2: N/A

3. Have the authors made all data underlying the findings in their manuscript fully available?

Reviewer #1: Yes

Reviewer #2: No

4. Is the manuscript presented in an intelligible fashion and written in standard English?

Reviewer #1: Yes

Reviewer #2: Yes

5. Review Comments to the Author

Reviewer #1: Comments to the Author

Congratulations on the submitted manuscript. The topic is timely and will be of interest to the readers of the journal. However, a few changes are suggested to improve the clarity of this manuscript. I have several recommendations and questions about the manuscript.

Abstract

A descriptive qualitative design informed this study. We conducted focus groups with a range of interprofessional healthcare providers, administrators, and staff working in PHC teams

across Ontario. Eight focus groups were conducted with 39 participants representing the six

health regions of the province.

-Harmonize the statements:

A qualitative approach was used for this study, which involved 39 participants from six health regions of the province. Eight focus groups were conducted----

Introduction

-Very good explanation.

Methodology

Our study used a qualitative description approach, using focus groups to explore the

experiences of participants.

-Here need to explain how many participants were involved in this study.

Data collection and data analysis took place simultaneously, with data analysis following

the thematic analysis process as outlined by Braun & Clark [29].

-Explain the steps of data analysis using Braun & Clark(2006), to make it more clear for the readers.

Discussion

-Very good.

References

-Most of the citations are above 5 years. Very good.

Reviewer #2: Overall, this is a well-executed study on an important topic, written up clearly and with a nice selection of meaningful excerpts.

I do have one larger overarching concern and several smaller issues that are related.

Major concern:

1. The research question is appropriately stated as " to identify the facilitators and challenges *experienced* by interprofessional PHC team" (my highlight). Understanding the experiences of key players in the healthcare system is important, but it is also important to be mindful of the limitation that study design brings with it. For example, both the analysis and the conclusions highlights the importance of a trusting relationship between PHC provider and patient to combat vaccine hesitancy. But while this may be the case (and there's other, cited literature to support the idea), this study provides virtually no data that could actually support this as a finding: we don't know if patients actually valued the relationship or if they were just being polite, if some patients would actually prefer a non-PHC setting, if the providers saw a biased sample of patients, etc. More generally, based on the study design, we simply don't stand to learn much about what PHCs are (comparatively) good at in the context of vaccinations -- all we have is their self-assessment. One way to think about this might be that observations about *process* are more likely to be analytically relevant then observations about *outcomes*.

What we *can* (and do) learn, on the other hand, are how they operated, where they perceived challenges, how they solved problems: accordingly the sections about red tape or lack of inclusion in systemic planning are both the most interesting and the most analytically relevant sections of the paper.

I'd encourage authors to

- expand on those themes at the expense of some of the themes that provide less analytical leverage, *especially* in the discussion section of the paper

- carefully weigh, phrase, and potentially omit specific policy conclusions based on the type of evidence they're based on.

Minor concerns:

2. As noted above, the discussion on the lack of inclusion in systemic planning is particularly strong and fairly long. I wonder if authors' coding would allow dividing this into two more fine-grained codes to strengthen the precision of the analysis and the clarity of presentation.

3. In the discussion, the paper notes in several places variation between different PHCs (e..g. on vaccine access, handling red tape, etc.). This variation is very interesting, but does not come through much in the analysis section. I'd like to see that explored more clearly, including more on the nature/drivers of variation to the extent those can be discerned.

4. The discussion section overall is a bit long and unfocussed, going through every theme one-by-one. I'd encourage focussing this more and picking some highlights in the findings for more detailed discussion, leaving other topics out.

5. The conclusion seems a bit anticlimactic and short. Taking up one or two key issues, highlighting how the study has improved our understanding and suggesting further avenues for study would all be nice to see there.

6. Data availability: I understand that data may not be shareable given consent and REB protocol, but if it can be made available on reasonable request, it should be made available through a data repository as per PLOS's data policy. If that's not possible, or data can only be shared by permission from the REB or similar, that should be clearly stated. Data sharing by request to the author is not in line with PLOS's policy.

6. PLOS authors have the option to publish the peer review history of their article (what does this mean?). If published, this will include your full peer review and any attached files.

Reviewer #1: **Yes: **DR RUSNANI AB LATIF

Reviewer #2: **Yes: **Sebastian Karcher

---

## [Author Response · Author response to Decision Letter 0]

26 Jan 2024

Dear Dr. Deniz:

We are pleased to submit revisions to our manuscript titled “A Qualitative Examination of the Facilitators and Barriers Shaping Primary Health Care Teams’ Experiences in the Distribution of the COVID-19 Vaccination in Ontario, Canada”. We appreciate the reviewer’s feedback and have revised the manuscript in a way that responds to all editor and reviewer comments. Please find below an itemized response to all reviewer and editor comments: 

Reviewer 1 Comment

Congratulations on the submitted manuscript. The topic is timely and will be of interest to the readers of the journal. 

Response:

Thank you! We appreciate your enthusiasm and positive feedback. We also appreciate the time you took to review our manuscript. 

Reviewer 1 Comment:

Abstract: A descriptive qualitative design informed this study. We conducted focus groups with a range of interprofessional healthcare providers, administrators, and staff working in PHC teams

 across Ontario. Eight focus groups were conducted with 39 participants representing the six health regions of the province.

 -Harmonize the statements: A qualitative approach was used for this study, which involved 39 participants from six health regions of the province. Eight focus groups were conducted----

Response:

We revised the methods section of abstract which now reads: 

“A qualitative approach was used for this study, which involved 39 participants from the six health regions of the province. Eight focus groups were conducted with a range of interprofessional healthcare providers, administrators, and staff working in PHC teams across Ontario. The sample reflected a diverse range of clinical, administrative, and leadership roles in PHC. Focus groups were audio-recorded and transcribed, while transcriptions were then analyzed using thematic analysis.”

Reviewer 1 Comment

Introduction

 -Very good explanation. 

Response: 

Thank you!

Reviewer 1 Comment:

Methodology

Our study used a qualitative description approach, using focus groups to explore the

 experiences of participants.

 -Here need to explain how many participants were involved in this study. 

Response: 

The sentence now reads as: 

“Our study used a qualitative description approach, using focus groups to explore the experiences and perspectives of 39 participants.”

Reviewer 1 Comment:

Data collection and data analysis took place simultaneously, with data analysis following the thematic analysis process as outlined by Braun & Clark [29].

-Explain the steps of data analysis using Braun & Clark(2006), to make it more clear for the readers. 

Response: 

We added the following sentence early in the data analysis section: 

“The six steps as outlined by Braun and Clarke [29] included i) familiarization with the data, ii) generating initial codes, iii) searching for themes, iv) reviewing the themes, v) defining and naming the themes, and vi) producing a report.”

The following sentence was also added to the data analysis section: 

“The primary and secondary data analysts familiarized themselves by reading transcripts prior to coding.”

Reviewer 1 Comment:

Discussion

 -Very good.

 References

 -Most of the citations are above 5 years. Very good. 

Response: 

Thank you!

Reviewer 2 Comment 

Overall, this is a well-executed study on an important topic, written up clearly and with a nice selection of meaningful excerpts. 

Response: 

Thank you! We appreciate your positive feedback and the time you took to review our manuscript. 

Reviewer 2 Comment:

The research question is appropriately stated as " to identify the facilitators and challenges *experienced* by interprofessional PHC team" (my highlight). 

Response: 

We revised the objective which now reads as: “The key objective informing this study was to explore the experiences and perspectives of interprofessional PHC teams in the distribution of COVID-19 vaccination across Ontario.” We believe that the revised objective is better aligned with the study aim and the results. 

We revised the title to: 

“A qualitative examination of the experiences and perspectives of interprofessional primary health care teams in the distribution of the COVID-19 vaccination in Ontario, Canada”

Reviewer 2 Comment:

Understanding the experiences of key players in the healthcare system is important, but it is also important to be mindful of the limitation that study design brings with it. For example, both the analysis and the conclusions highlights the importance of a trusting relationship between PHC provider and patient to combat vaccine hesitancy. But while this may be the case (and there's other, cited literature to support the idea), this study provides virtually no data that could actually support this as a finding: we don't know if patients actually valued the relationship or if they were just being polite, if some patients would actually prefer a non-PHC setting, if the providers saw a biased sample of patients, etc. More generally, based on the study design, we simply don't stand to learn much about what PHCs are (comparatively) good at in the context of vaccinations -- all we have is their self-assessment. 

Response: 

Thank you for noting this important point. The following sentences are added to the limitation section: 

“Although understanding the experiences of key players in the healthcare system is important, it is important to recognize that the lack of patients’ perspectives is a limitation in our study. Healthcare providers in our study overwhelmingly emphasized the importance and role of trusting relationships in the distribution of the COVID-19 vaccination yet without the perspective of patients, we cannot confirm the accuracy of healthcare providers’ perspectives given the dynamic nature of therapeutic relationships. Despite this limitation, we believe our study provides unique insights about the experiences and perceptions of interprofessional providers and PHC teams.”

Reviewer 2 Comment:

One way to think about this might be that observations about *process* are more likely to be analytically relevant then observations about *outcomes*. What we *can* (and do) learn, on the other hand, are how they operated, where they perceived challenges, how they solved problems: accordingly the sections about red tape or lack of inclusion in systemic planning are both the most interesting and the most analytically relevant sections of the paper. I'd encourage authors to expand on those themes at the expense of some of the themes that provide less analytical leverage, *especially* in the discussion section of the paper; carefully weigh, phrase, and potentially omit specific policy conclusions based on the type of evidence they're based on. 

Response: 

We revised the objective which now reads as: “The key objective informing this study was to explore the experiences and perspectives of interprofessional PHC teams in the distribution of COVID-19 vaccination across Ontario.” We believe that the revised objective is better aligned with the study aim and the results and addresses some of the concerns raised.

Reviewer 2 Comment:

As noted above, the discussion on the lack of inclusion in systemic planning is particularly strong and fairly long. I wonder if authors' coding would allow dividing this into two more fine-grained codes to strengthen the precision of the analysis and the clarity of presentation. 

Response: 

Thank you for the suggestion. After re-reviewing the results and going back into the data, we have opted not to re-code the data as we believe that the current codes and grouping reflect what we heard from the participants. 

Reviewer 2 Comment:

In the discussion, the paper notes in several places variation between different PHCs (e..g. on vaccine access, handling red tape, etc.). This variation is very interesting, but does not come through much in the analysis section. I'd like to see that explored more clearly, including more on the nature/drivers of variation to the extent those can be discerned. 

Response: 

The following sentence is added to the discussion: “Some of these variations may be because of differences inherent in the community context, patient population, and/or organizational structures such as team composition, resourcing, and leadership [31, 32].”

Reviewer 2 Comment:

The discussion section overall is a bit long and unfocussed, going through every theme one-by-one. I'd encourage focussing this more and picking some highlights in the findings for more detailed discussion, leaving other topics out. 

Response: 

The discussion section has been substantively edited for length and focus. 

Reviewer 2 Comment:

The conclusion seems a bit anticlimactic and short. Taking up one or two key issues, highlighting how the study has improved our understanding and suggesting further avenues for study would all be nice to see there. 

Response: 

The conclusion has been revised and now reads: 

“PHC teams were instrumental in supporting COVID-19 vaccinations in Ontario. Because of the longstanding experience in delivering vaccinations and diverse expertise, PHC teams were able to quickly mobilize and draw on the collective strengths of a breadth of professions. PHC teams need to be included in the planning and strategizing of future vaccination endeavors to optimize their capacity.”

Reviewer 2 Comment: 

Data availability: I understand that data may not be shareable given consent and REB protocol, but if it can be made available on reasonable request, it should be made available through a data repository as per PLOS's data policy. If that's not possible, or data can only be shared by permission from the REB or similar, that should be clearly stated. Data sharing by request to the author is not in line with PLOS's policy. 

Response: 

We are unable to publicly share our data set because although we have removed identifiable information from transcripts, we cannot remove all information that could potentially be used to identify participants. Responses to questions contain information describing Family Health Team communities, practice settings, and providers’ personal roles and responsibilities, some of whom live in small rural communities where such information could be used to identify individuals. Our REB requires that we not share identifiable information because of the potential for identifying participants. 

Editor Comment

Response: 

The manuscript was revised throughout to meet the PLOS ONE’s style requirements including headings, references, and title page. 

Editor Comment:

We note that you have indicated that data from this study are available upon request. PLOS only allows data to be available upon request if there are legal or ethical restrictions on sharing data publicly. For more information on unacceptable data access restrictions, please see http://journals.plos.org/plosone/s/data-availability#loc-unacceptable-data-access-restrictions.

Response: 

We are unable to publicly share our data set because although we have removed identifiable information from transcripts, we cannot remove all information that could potentially be used to identify participants. Responses to questions contain information describing Family Health Team communities, practice settings, and providers’ personal roles and responsibilities, some of whom live in small rural communities where such information could be used to identify individuals. 

Editor Comment:

Your ethics statement should only appear in the Methods section of your manuscript. If your ethics statement is written in any section besides the Methods, please delete it from any other section. 

Response: 

The ethics statement is located in the Methods section.

---

## [Editor Report · Decision Letter 1]

7 Feb 2024

PONE-D-23-23366R1A qualitative examination of the experiences and perspectives of interprofessional primary health care teams in the distribution of the COVID-19 vaccination in Ontario, CanadaPLOS ONE

Dear Dr. Ashcroft,

Thank you for submitting your manuscript to PLOS ONE. After careful consideration, we feel that it has merit but does not fully meet PLOS ONE’s publication criteria as it currently stands. Therefore, we invite you to submit a revised version of the manuscript that addresses the points raised during the review process.

We look forward to receiving your revised manuscript.

Kind regards,

Elif Ulutaş Deniz

Academic Editor

PLOS ONE

---

## [Author Response · Author response to Decision Letter 1]

8 Feb 2024

Dr. Elif Ulutaş Deniz 

Academic Editor 

PLOS One 

January 02, 2024 

Dear Dr. Deniz: 

We are pleased to submit revisions to our manuscript titled “A Qualitative Examination of the Facilitators and Barriers Shaping Primary Health Care Teams’ Experiences in the Distribution of the COVID-19 Vaccination in Ontario, Canada”. We appreciate the reviewer’s feedback and have revised the manuscript in a way that responds to all editor and reviewer comments. Please find below an itemized response to all reviewer and editor comments: 

Reviewer 1 Comment:

Congratulations on the submitted manuscript. The topic is timely and will be of interest to the readers of the journal. 

Response: 

Thank you! We appreciate your enthusiasm and positive feedback. We also appreciate the time you took to review our manuscript. 

Reviewer 1 Comment: 

Abstract: A descriptive qualitative design informed this study. We conducted focus groups with a range of interprofessional healthcare providers, administrators, and staff working in PHC teams 

 across Ontario. Eight focus groups were conducted with 39 participants representing the six health regions of the province. Harmonize the statements: A qualitative approach was used for this study, which involved 39 participants from six health regions of the province. Eight focus groups were conducted---- 

Response: 

We revised the methods section of abstract which now reads: 

Reviewer 1 Comment:

“A qualitative approach was used for this study, which involved 39 participants from the six health regions of the province. Eight focus groups were conducted with a range of interprofessional healthcare providers, administrators, and staff working in PHC teams across Ontario. The sample reflected a diverse range of clinical, administrative, and leadership roles in PHC. Focus groups were audio-recorded and transcribed, while transcriptions were then analyzed using thematic analysis.” 

Introduction: Very good explanation. 

Response: 

Thank you! 

Reviewer 1: 

Methodology: Our study used a qualitative description approach, using focus groups to explore the 

 experiences of participants. 

 -Here need to explain how many participants were involved in this study. 

Response: 

The sentence now reads as: 

“Our study used a qualitative description approach, using focus groups to explore the experiences and perspectives of 39 participants.” 

Reviewer 1 Comment:

Data collection and data analysis took place simultaneously, with data analysis following 

 the thematic analysis process as outlined by Braun & Clark [29]. 

 -Explain the steps of data analysis using Braun & Clark(2006), to make it more clear for the readers. 

Response:

We added the following sentence early in the data analysis section: 

“The six steps as outlined by Braun and Clarke [29] included i) familiarization with the data, ii) generating initial codes, iii) searching for themes, iv) reviewing the themes, v) defining and naming the themes, and vi) producing a report.” 

The following sentence was also added to the data analysis section: “The primary and secondary data analysts familiarized themselves by reading transcripts prior to coding.” 

Reviewer 1 Comment:

Discussion 

 -Very good. 

 References 

 -Most of the citations are above 5 years. Very good. 

Response: Thank you! 

Reviewer 2 Comment:

Overall, this is a well-executed study on an important topic, written up clearly and with a nice selection of meaningful excerpts. 

Response:

Thank you! We appreciate your positive feedback and the time you took to review our manuscript. 

Reviewer 2 Comment:

The research question is appropriately stated as " to identify the facilitators and challenges *experienced* by interprofessional PHC team" (my highlight). 

Response: 

We revised the objective which now reads as: “The key objective informing this study was to explore the experiences and perspectives of interprofessional PHC teams in the distribution of COVID-19 vaccination across Ontario.” We believe that the revised objective is better aligned with the study aim and the results. 

We revised the title to: 

“A qualitative examination of the experiences and perspectives of interprofessional primary health care teams in the distribution of the COVID-19 vaccination in Ontario, Canada” 

Reviewer 2 Comment:

Understanding the experiences of key players in the healthcare system is important, but it is also important to be mindful of the limitation that study design brings with it. For example, both the analysis and the conclusions highlights the importance of a trusting relationship between PHC provider and patient to combat vaccine hesitancy. But while this may be the case (and there's other, cited literature to support the idea), this study provides virtually no data that could actually support this as a finding: we don't know if patients actually valued the relationship or if they were just being polite, if some patients would actually prefer a non-PHC setting, if the providers saw a biased sample of patients, etc. More generally, based on the study design, we simply don't stand to learn much about what PHCs are (comparatively) good at in the context of vaccinations -- all we have is their self-assessment. 

Response: 

Thank you for noting this important point. The following sentences are added to the limitation section: 

“Although understanding the experiences of key players in the healthcare system is important, it is important to recognize that the lack of patients’ perspectives is a limitation in our study. Healthcare providers in our study overwhelmingly emphasized the importance and role of trusting relationships in the distribution of the COVID-19 vaccination yet without the perspective of patients, we cannot confirm the accuracy of healthcare providers’ perspectives given the dynamic nature of therapeutic relationships. Despite this limitation, we believe our study provides unique insights about the experiences and perceptions of interprofessional providers and PHC teams.” 

Reviewer 2 Comment:

One way to think about this might be that observations about *process* are more likely to be analytically relevant then observations about *outcomes*. What we *can* (and do) learn, on the other hand, are how they operated, where they perceived challenges, how they solved problems: accordingly the sections about red tape or lack of inclusion in systemic planning are both the most interesting and the most analytically relevant sections of the paper. I'd encourage authors to expand on those themes at the expense of some of the themes that provide less analytical leverage, *especially* in the discussion section of the paper; carefully weigh, phrase, and potentially omit specific policy conclusions based on the type of evidence they're based on. 

Response: 

We revised the objective which now reads as: “The key objective informing this study was to explore the experiences and perspectives of interprofessional PHC teams in the distribution of COVID-19 vaccination across Ontario.” We believe that the revised objective is better aligned with the study aim and the results and addresses some of the concerns raised. 

Reviewer 2 Comment:

As noted above, the discussion on the lack of inclusion in systemic planning is particularly strong and fairly long. I wonder if authors' coding would allow dividing this into two more fine-grained codes to strengthen the precision of the analysis and the clarity of presentation. 

Response:

Thank you for the suggestion. After re-reviewing the results and going back into the data we have opted not to re-code the data as we believe that the current codes and grouping reflect what we heard from the participants. 

Reviewer 2 Comment:

In the discussion, the paper notes in several places variation between different PHCs (e..g. on vaccine access, handling red tape, etc.). This variation is very interesting, but does not come through much in the analysis section. I'd like to see that explored more clearly, including more on the nature/drivers of variation to the extent those can be discerned. 

Response:

The following sentence is added to the discussion: “Some of these variations may be because of differences inherent in the community context, patient population, and/or organizational structures such as team composition, resourcing, and leadership [31, 32].” 

Reviewer 2 Comment:

The discussion section overall is a bit long and unfocussed, going through every theme one-by-one. I'd encourage focussing this more and picking some highlights in the findings for more detailed discussion, leaving other topics out. 

Response:

The discussion section has been substantively edited for length and focus. 

Reviewer 2 Comment:

The conclusion seems a bit anticlimactic and short. Taking up one or two key issues, highlighting how the study has improved our understanding and suggesting further avenues for study would all be nice to see there. 

Response:

The conclusion has been revised and now reads: 

“PHC teams were instrumental in supporting COVID-19 vaccinations in Ontario. Because of the longstanding experience in delivering vaccinations and diverse expertise, PHC teams were able to quickly mobilize and draw on the collective strengths of a breadth of professions. PHC teams need to be included in the planning and strategizing of future vaccination endeavors to optimize their capacity.” 

Reviewer 2 Comment:

Data availability: I understand that data may not be shareable given consent and REB protocol, but if it can be made available on reasonable request, it should be made available through a data repository as per PLOS's data policy. If that's not possible, or data can only be shared by permission from the REB or similar, that should be clearly stated. Data sharing by request to the author is not in line with PLOS's policy. 

Response:

We are unable to publicly share our data set because although we have removed identifiable information from transcripts, we cannot remove all information that could potentially be used to identify participants. Responses to questions contain information describing Family Health Team communities, practice settings, and providers’ personal roles and responsibilities, some of whom live in small rural communities where such information could be used to identify individuals. Our REB requires that we not share identifiable information because of the potential for identifying participants.

---

## [Decision Letter · Decision Letter 2]

15 May 2024

A qualitative examination of the experiences and perspectives of interprofessional primary health care teams in the distribution of the COVID-19 vaccination in Ontario, Canada

PONE-D-23-23366R2

Dear Dr. Ashcroft,

We’re pleased to inform you that your manuscript has been judged scientifically suitable for publication and will be formally accepted for publication once it meets all outstanding technical requirements.

Kind regards,

Elif Ulutaş Deniz

Academic Editor

PLOS ONE

Additional Editor Comments (optional):

Reviewers' comments:

Reviewer's Responses to Questions

**Comments to the Author**

1. If the authors have adequately addressed your comments raised in a previous round of review and you feel that this manuscript is now acceptable for publication, you may indicate that here to bypass the “Comments to the Author” section, enter your conflict of interest statement in the “Confidential to Editor” section, and submit your "Accept" recommendation.

Reviewer #1: All comments have been addressed

Reviewer #2: All comments have been addressed

2. Is the manuscript technically sound, and do the data support the conclusions?

Reviewer #1: Yes

Reviewer #2: Yes

3. Has the statistical analysis been performed appropriately and rigorously? 

Reviewer #1: Yes

Reviewer #2: N/A

4. Have the authors made all data underlying the findings in their manuscript fully available?

Reviewer #1: Yes

Reviewer #2: No

5. Is the manuscript presented in an intelligible fashion and written in standard English?

Reviewer #1: Yes

Reviewer #2: Yes

6. Review Comments to the Author

Reviewer #1: Dear Authors

Congratulations on your manuscript, I have checked the reviewer’s comment, and all the amendments were implemented in response to the reviewers' recommendations and commendations.

Reviewer #2: Thank you for a much more nuanced discussion section, also reflected in the changed title. Since some fairly strong statements that suggest generalizability remain in the manuscript, I'd suggest listing some additional limitations related to sample size and recruitment in the limitation section, but will leave this to the authors.

7. PLOS authors have the option to publish the peer review history of their article (what does this mean?). If published, this will include your full peer review and any attached files.

Reviewer #1: **Yes: **DR RUSNANI AB LATIF

Reviewer #2: **Yes: **Sebastian Karcher

---

## [Editor Report · Acceptance letter]

20 May 2024

PONE-D-23-23366R2 

PLOS ONE

Dear Dr. Ashcroft, 

I'm pleased to inform you that your manuscript has been deemed suitable for publication in PLOS ONE. Congratulations! Your manuscript is now being handed over to our production team.

Kind regards, 

on behalf of

Dr. Elif Ulutaş Deniz 

Academic Editor

PLOS ONE